# The Dual Role of Autophagy in Postischemic Brain Neurodegeneration of Alzheimer’s Disease Proteinopathy

**DOI:** 10.3390/ijms241813793

**Published:** 2023-09-07

**Authors:** Ryszard Pluta

**Affiliations:** Department of Pathophysiology, Medical University of Lublin, 20-090 Lublin, Poland; pluta2018@wp.pl

**Keywords:** brain ischemia, reperfusion, recirculation, Alzheimer’s disease proteinopathy, autophagy, necrosis, apoptosis, neurons, glial cells, endothelial cells, neurodegeneration

## Abstract

Autophagy is a self-defense and self-degrading intracellular system involved in the recycling and elimination of the payload of cytoplasmic redundant components, aggregated or misfolded proteins and intracellular pathogens to maintain cell homeostasis and physiological function. Autophagy is activated in response to metabolic stress or starvation to maintain homeostasis in cells by updating organelles and dysfunctional proteins. In neurodegenerative diseases, such as cerebral ischemia, autophagy is disturbed, e.g., as a result of the pathological accumulation of proteins associated with Alzheimer’s disease and their structural changes. Postischemic brain neurodegeneration, such as Alzheimer’s disease, is characterized by the accumulation of amyloid and tau protein. After cerebral ischemia, autophagy was found to be activated in neuronal, glial and vascular cells. Some studies have shown the protective properties of autophagy in postischemic brain, while other studies have shown completely opposite properties. Thus, autophagy is now presented as a double-edged sword with possible therapeutic potential in brain ischemia. The exact role and regulatory pathways of autophagy that are involved in cerebral ischemia have not been conclusively elucidated. This review aims to provide a comprehensive look at the advances in the study of autophagy behavior in neuronal, glial and vascular cells for ischemic brain injury. In addition, the importance of autophagy in neurodegeneration after cerebral ischemia has been highlighted. The review also presents the possibility of modulating the autophagy machinery through various compounds on the development of neurodegeneration after cerebral ischemia.

## 1. Introduction

### 1.1. Epidemiology of Brain Ischemia

Ischemic brain injury in humans develops as a result of a sudden partial or complete occlusion of the cerebrovascular network supplying blood to the brain [1]. Brain ischemia can occur in individuals with immature and mature brains. Perinatal ischemic stroke occurs between the 20th week of pregnancy and the 28th day after birth [2,3]. The incidence of perinatal stroke is 29 per 100,000 live births per year [2,3,4,5]. Despite current therapies, at least 1 in 10 children after the first ischemic stroke have a recurrence in the next 5 years [2]. The annual direct cost of stroke in children in the US, counting inpatient and outpatient services, is approximately USD 1,000,000 [2]. Perinatal ischemic brain injury is not only the leading cause of mortality in the early days of life, but also neonates who survive and develop neurological disabilities, cognitive deficits and behavioral impairments that often last a lifetime, such as in the form of dementia [3].

It is estimated that ischemic focal brain injury in adults, which accounts for roughly 85–90% of all cases, is the dominant cause of progressive and irreversible disability in humans and the second cause of death [6,7,8,9,10,11,12,13,14]. The incidence of ischemic brain alterations increases with age in developed and developing countries, with the exception of China and India, where the incidence of brain ischemia has increased sharply in people under 40 years of age [15]. As of 2015, focal ischemic brain injury is the leading cause of death and disability in China, posing a very serious threat to the health of the country’s citizens [16]. According to China’s official brain ischemia program, it was estimated that 17.8 million Chinese citizens had a stroke in 2020, of which 3.4 million had a first-ever stroke and another 2.3 million died from a stroke [16]. In addition, roughly 12.5% of stroke survivors remain disabled for life, equivalent to 2.2 million stroke-associated disabilities in 2020 [16]. It was calculated that the cost of hospitalization due to stroke in 2020 was CNY 58.0 billion, of which patients paid CNY 19.8 billion [16]. These figures are staggering considering that China makes up only 18% of the world’s population [17]. In fact, stroke incidence and mortality in China are 28% and 35% higher than the global average, respectively [17]. In addition, China’s estimated lifetime risk of stroke is 39.3% for people aged 25 and over, significantly higher than the global average of 24.9% [17].

A meta-analysis of papers on ischemic stroke in India showed that in 18 analyzed papers in five studies, the age of patients was below 40 years [18]. The age range of the patients was between 32 and 67 years, with a mean of 54 ± 9 [18]. In the study, 64% was men [18]. Countries such as India estimate that approximately 14% of global disability-adjusted life years have been lost due to stroke [19]. It has been documented that 50–70% of stroke survivors regain independence, but 15–30% are permanently disabled and 20% require institutional care, including 3 months after the onset of the stroke [19]. This worrying trend shows that the rate of stroke among people aged 20 to 54 worldwide has increased from 13% in 1990 to 19% in 2016 [20,21]. In the 21st century, the number of cases of ischemic stroke in young adults has increased to approximately two million per year [22].

It should be emphasized that, currently, 70% of ischemic strokes and 87% of related deaths and irreversible disabilities predominant occur in poor countries [23]. In poor countries, the number of cases of ischemic stroke has more than doubled over the last forty years, occurring approximately 15 years earlier and causing more deaths than in developed countries [23,24]. Approximately 84% of stroke case patients in poor countries die within three years, compared with only 16% in developed countries [23]. Currently, it is estimated that there are approximately 15 million cases of ischemic stroke annually, of which approximately half die within a year [1,6,7,9,10,11,14]. It is also known that the number of postischemic patients across the world has reached approximately 33 million [10]. Due to aging in the European community, it is estimated that by 2025, the incidence of stroke in this group is likely to increase to 1.5 million [22]. The number of stroke survivors in the European Union is estimated to increase by 27% by 2047 due to an aging society and higher survival rates of the population [25].

Over the past 25 years, there has been a decrease in the death rate of ischemic stroke survivors around the world, despite an increase in the number of cases as a result of increased life expectancy [12]. In rich countries, cerebral ischemia occurs 10 times more often than hemorrhagic stroke, while in poor countries, the advantage is definitely smaller [12]. The risk of stroke recurrence after a first ischemic stroke in the first month of treatment has been shown to be high, at 1 in 25 [12]. The data indicate that individuals who have had a stroke have a high chance of having another stroke in the first year of approximately 10% and annually in subsequent years of approximately 5% [26]. Symptoms usually depend on the extent of ischemia and the region of the brain involved, and include sensory and motor disturbances that are generally permanent. Over 30–50% of people who have experienced cerebral ischemia are functionally dependent on other people [8]. One year after a stroke, 10% to 15% of survivors require assistance in a specialist facility [27]. Epidemiological studies indicate that the incidence of stroke in middle-aged people, i.e., 50–70 years old, is higher compared to people over 70 years old [28]. Stroke patients aged >85 years account for 17% of all cases, and in this age group, stroke is more common in women than in men [2]. On the other hand, older patients show a marked decline in overall performance between 6 and 30 months poststroke [29]. According to recent projections, by 2030, the number of stroke survivors is predicted increase to 77 million [10,30,31]. Despite this, the global death rate due to cerebral ischemia has decreased significantly [12]. If these global stroke trends continue, by 2030 there is predicted to be approximately 70 million stroke cases, approximately 12 million deaths and more than 200 million disability-adjusted life years annually [30,31]. Despite significant progress in the diagnosis and treatment of brain ischemia, it is assumed that around 2050, the number of brain ischemia cases is likely to double, and disability after stroke is predicted to also increase due to the increasing number of patients whom are likely to survive an ischemic episode [32,33].

### 1.2. Medical, Financial and Social Burdens of Brain Ischemia

Therefore, it is no surprise that the socioeconomic impact of stroke worldwide is massive and growing over time; for example, the annual cost in the European Union was EUR 38 billion in 2012, EUR 45 billion in 2015 and EUR 60 billion in 2017 [34]. In addition, the additional cost of treating people with postischemic dementia and other dementias is estimated to increase from USD 321 billion in 2022 to nearly USD 1 trillion by 2050 [35]. On top of this, patients suffering from poststroke dementia are more likely to develop various complications and numerous chronic diseases, which could certainly generate an even greater financial burden related to the treatment of comorbidities; hence, brain ischemia has become a global health problem [36]. Finally, human ischemic stroke is associated with a very low cure rate, negligible full recovery, frequent recurrence, permanent disability and high mortality [37].

### 1.3. Postischemic Neurodegeneration

Cerebral ischemia has been found to trigger a sequence of pathological events that can last from minutes to the remaining years of life [8,38,39,40,41]. These pathologies include energy failure, oxidative stress, excitotoxicity, neuroinflammation, cortical and subcortical infarcts, white matter rarefaction, blood–brain barrier damage, microbleeding and cerebral amyloid angiopathy [8,14,30,31,42,43,44,45]. The consequence of the above changes is additional hypoperfusion, causing the ischemia of the adjacent areas, gliosis, the accumulation of amyloid plaques and neurofibrillary tangles, neuronal death and, ultimately, brain atrophy [38,39,41,46,47,48,49]. A focal ischemic episode typically damages the brain cortex, hippocampus, temporal lobe, entorhinal cortex, amygdala and parahippocampus to varying degrees. Postischemia, these structures are involved in cognitive and memory deficits, and their progressive degeneration also induces behavioral changes. Cognitive impairment due to ischemia is mild to severe, and has been seen in approximately 35–70% of survivors one year after a stroke [50,51,52,53,54,55]. It should be noted, however, that it is common for cognitive function to fail to return to before stroke levels [55,56,57,58]. Dementia has been shown to occur even in cases with transient cognitive impairment after ischemia [55,59]. Studies show that cerebral ischemia accelerates the onset of dementia by approximately 10 years [60]. It is estimated that 8–13% of patients develop dementia immediately after a first stroke, and over 40% after a second stroke [10,54,60]. The estimated progression of dementia in patients who survive 25 years after a stroke is approximately 48% [46,60].

It is believed that one in six people in the world are likely to suffer a brain ischemia in their lifetime [1]. This is accompanied by massive neuronal death in vulnerable brain regions. Thus, understanding the molecular mechanisms of neuronal death arising from different forms of ischemic insults is a major goal of investigators in the field. Thus far, three pathways of ischemic neuronal cell death, such as necrotic, apoptotic and autophagocytotic, have been identified. Postischemic brain neurodegeneration, such as that seen in Alzheimer’s disease, is a progressive neurodegenerative disease with two progressive pathological changes, i.e., extracellular amyloid plaques composed of β-amyloid peptide and intracellular neurofibrillary tangles composed of hyperphosphorylated tau protein. At present, there are no sufficient therapeutic strategies available. As there are currently no neuroprotective substances known to exist, neuroprotective molecular mechanisms have not been explained to this day. In contrast to necrosis and apoptosis, autophagy could possibly serve as a potential therapeutic target against ischemia–reperfusion brain injury [61].

Over the past two decades, it has been proposed that ischemic neuronal death is associated with folding molecules, such as amyloid and tau protein. Postischemia is characterized by progressive memory loss and cognitive impairment, which, finally, ends in full-blown dementia with the Alzheimer’s disease phenotype. Given the facts presented, postischemic therapy strategies should probably focus on two characteristic changes: pathogenic misfolded amyloid and tau protein with the hope of affecting macroautophagy, also called autophagy.

### 1.4. Autophagy as Hope after an Ischemic Episode

There are three types of autophagy in mammalian cells: macroautophagy, microautophagy and chaperon-mediated autophagy. Macroautophagy is the best-studied type and a widely recognized autophagy in mammalian cells; thus, this review focuses on macroautophagy [61,62,63,64,65,66,67,68,69,70]. The process of the development of macroautophagy, hereinafter referred to as “autophagy”, consists of a series of successive stages [67,69,70]. The first is the creation of a phagophore. After autophagy-inducing signals appear, a small liposome-like membrane structure forms somewhere in the cytoplasm. The membrane then continues to expand to form a flat lipid bilayer called the phagophore, which is a form of direct evidence in the initiation of autophagy. In the second stage, the autophagosome is formed. To this end, the phagophore is constantly stretched to incorporate various components and, finally, transforms into a spherical double-membrane structure, namely, the autophagosome [61,67,69,70], which randomly or selectively captures misfolded proteins or damaged organelles, for example, misfolded tau protein or amyloid [62,69,70]. Regarding autophagosomal membrane elongation, various ATG vesicles and ubiquitin-like binding systems are involved in this phenomenon [67,69,70]. Autophagosomes then fuse with lysosomes to form autophagolysosomes [67,69,70]. Finally, the autophagy cargo is broken down by lysosomal enzymes and the recovered nutrients, including amino acids, fatty acids, etc., are transported back into the cytoplasm as part of the recycling mechanism, while residues are excreted from the cell to the outside. Recently, autophagy has been shown to involve a wide range of signal regulation pathways, which have mainly been divided into the mammalian target of rapamycin-dependent and the mammalian target of rapamycin-independent pathways, creating a sophisticated and intricate network of signals that regulate autophagy either positively or negatively [61,67,69,70].

It has been widely accepted that autophagy is the self-defense of the cellular catabolic pathway through which some long-lived or misfolded proteins and damaged organelles are broken down into metabolic substances and recycled to maintain cellular homeostasis [62,67,69,70]. During the process of autophagy, dysfunctional and unnecessary proteins and cellular elements are surrounded with a double-membrane vesicle called the autophagosome and, next, the autophagosome fuses with the lysosome, which, ultimately, leads to the recycling and degradation of redundant intracellular structures and proteins [62,67,69,70]. Autophagy is very important for cell and tissue homeostasis and is actively involved in the aging process, as well as in many human and animal diseases, including neurodegenerative diseases such as Alzheimer’s disease or postischemic neurodegeneration [62,63,64,65,66,67,68,69,70].

In experimental cerebral ischemia, autophagy has been shown to have a protective effect through the inhibition of neuronal apoptosis [71,72,73]. It has been shown that autophagy can be a double-edged sword in damage after cerebral ischemia; hence, it can be destructive or protective [61]. Thus, if the protective effects of autophagy can be controlled, autophagy can be a valuable therapeutic target, but if it cannot be controlled, it can be a messenger of death. It seems that the induction of autophagy could become a potential therapeutic strategy in the treatment of various diseases, including postischemic neurodegeneration. On the other hand, some scientists suggest that the overinduction of autophagy can lead to cell death, so-called autophagy cell death, emphasizing that the induction of autophagy in the treatment of diseases is not without complications. Further research on this topic is required to avoid such problems. We believe that there is likely to be a flexible adaptive capacity in different cells that face endogenous and exogenous stress. In a physiological situation, autophagy is activated after stress and helps cells survive by controlling the reuse and removal of dangerous intracellular cargo. In the above situation, autophagy causes a number of repair phenomena in the cells and even leads to the achievement of internal homeostasis by the cells, which results in a normal state. In contrast, if autophagy is impaired by pathogens and autophagy gene mutations, the adaptive capacity of cells decreases and cells are more susceptible to stress. On the other hand, if prolonged stress results in excessive or prolonged autophagy that exceeds the adaptive capacity of the cell, the overinduction of autophagy may trigger necrosis and apoptosis, ultimately, leading to cell death. Thus, autophagy appears to be a double-edged sword in the phenomenon of cellular adaptive machinery [74]. Whether autophagy is beneficial or harmful is determined by the rate of autophagy induction and the duration of its activation [67]. However, the role of autophagy in these processes is completely unclear and information about it in the literature is very limited. However, many different factors influence the occurrence and progression of cerebral ischemia. Although a great challenge has been undertaken to better understand postischemic brain neurodegeneration, some questions remain unanswered. Over the past two decades, increasing evidence has accumulated showing that autophagy is involved in the development of cerebral ischemic sequelae. To understand the contradictory findings presented above, it is important to pay attention to the level of autophagy, as too high or too low a level of activity can be harmful. The identification of the pathways that affect the balanced autophagy system could be of key importance in the development of therapies for diseases of the nervous system, including the postischemic neurodegeneration of Alzheimer’s disease proteinopathy. In this review, we present recent knowledge about the control of autophagy and its specific role in brain ischemia–reperfusion injury, and focus on the mechanisms and neuropathological processes that regulate autophagy in postischemic brain. Although great efforts have been undertaken to improve the understanding of brain ischemia, there remain unanswered questions. Over the past two decades, accumulating evidence has demonstrated that autophagy is extensively involved in brain ischemia. However, the exact role and molecular mechanisms of the autophagy process in ischemic insults are not fully elucidated. In this review, we provide a comprehensive overview of the advance in this exciting research field.

## 2. Search of the Literature

A search of the literature was performed using the following databases: Scopus, PubMed, Web of Science and Google Scholar. The keywords used in the article quest were cerebral ischemia, postischemic neurodegeneration, ischemic stroke, autophagy, neuronal death, necrosis, apoptosis, neuropathology and therapy, in various combinations. Articles from the databases had to be relevant and up-to-date, and only the most recent research was used in the review. The search focused mainly on articles published between 2000 and 2023. Previous original papers on the first descriptions of autophagy were also used.

## 3. Autophagy versus Postischemic Brain Cells

Brain ischemia is mainly caused by a cerebral blood flow blockage due to thrombosis or embolism, leading to an abnormal energy metabolism, sodium and chlorine influx, potassium efflux, cell membrane depolarization and cell edema. Subsequently, a series of damage cascades (calcium overload, excitatory amino acid toxicity, free radical generation, oxidative stress, neuroinflammation and apoptosis) can trigger irreversible brain damage and result in a positive feedback loop that, ultimately, causes severe damage to neuronal, glial and endothelial cells and their interconnections. In recent years, increasingly more studies have confirmed the important role of autophagy in the pathophysiological mechanisms of postischemic brain neurodegeneration [67,69]. Autophagy selectively targets dysfunctional organelles and intracellular microbes’ pathogenic proteins, and dysregulation in this process may lead to disease. In this review, we present the history of autophagy from the perspective of understanding and potentially reversing postischemic pathology in individual brain cells, as well as in the brain as a whole.

The first evidence that pyramidal neurons of the CA1 region of the hippocampus displayed membrane-bound vacuoles containing intracellular elements after brain ischemia with recirculation was documented in 1995 [75]. Additionally, cortical neurons presented cytoplasmic vacuoles connected with postischemic brain alterations [71,76,77].

The accumulation of autophagy-like vacuoles in astrocytes was identified in the brain following ischemic injury [78]. It was noted that autophagy was activated in ischemic astrocytes, which mildly decreased cell survival, but the braking of autophagy with 3-methyladenine significantly decreased the death of astrocytes after ischemic injury [79]. In contrast, the autophagy inhibitor 3-methyladenine increased hypoxic astrocyte death [80]. In the latest research, an explanation to the bidirectional activity of autophagy in astrocytes after ischemic brain injury was provided [81]. In brain hypoperfusion and focal brain ischemia, microglia cells activated the autophagy process [82,83]. The induction of microglial autophagy seemed to be associated with the deterioration of ischemia-induced neuronal cell injury [84]. The intensification of microglial autophagy activation was shown to be dependent on the time of both ischemia and reperfusion [85].

Increased autophagy-like cell death was observed in brain microvascular endothelial cells in p50 knockout mice with brain ischemia [86]. An enhancement in autophagy due to rapamycin has been demonstrated in ischemic brain microvascular endothelial cells, whereas the suppression of autophagy through 3-methyladenine intensified brain microvascular endothelial cell apoptosis [87], which indicated a protective effect of autophagy on brain microvascular endothelial cells and blood–brain barrier integrity after ischemic brain injury. Other data agree well with the fact that autophagy protects microvascular endothelial cells during ischemic stress, but autophagy inhibition through chloroquine increased the blood–brain barrier permeability and intensified brain edema [88]. These observations imply that the autophagy in the brain microvascular system triggered through ischemia may influence the outcome following ischemic brain injury. A postmortem investigation of human ischemic brain tissue presented a growth in staining in sequestosome 1 and the microtubule-associated protein 1 light chain 3, and the elevated appearance of autophagy vesicles following ischemic stroke [89]. These observations prove that changes in autophagy take place in animal and human brain tissue following brain ischemia, and imply that targeting autophagy after ischemia could be of clinical significance. Autophagy is an evolutionarily conserved process that involves the packaging, sequestration and delivery of used cytoplasm cargo to lysosomes, where lipids and proteins are degraded and recycled. Autophagy has been implicated in neuronal death in both acute and chronic neurodegenerative disorders. Autophagy in living beings is a homeostatic system for recycling organelles and proteins, and has been increasingly proposed as a treatment target for neurodegenerative disorders, including brain ischemia in humans called a stroke. A confirmation that the autophagy process occurs in the human ischemic brain is necessary before any therapies can begin in clinical trials. There seems to be no consensus regarding the role of autophagy in ischemic neuronal cells. Some studies have presented that neuronal death is related to activate autophagy.

## 4. Dysregulation of Autophagy Genes in Postischemic Brain

Evidence was provided for the lack of changes in the autophagy gene (*BECN 1*) during 2, 7 and 30 days after ischemia–reperfusion brain injury in hippocampal pyramidal neurons of the CA1 region [65,66]. These changes were accompanied by a huge overexpression of the *caspase 3* gene, responsible for apoptotic neuronal death [65,66] in neurons in the CA1 area. On the other hand, studies of the *BECN 1* gene expression in the CA3 area of the hippocampus showed a significant increase in its expression on the 30th day after ischemia. Parallel studies of the *CASP 3* gene in this region showed that this gene was significantly expressed between 7 and 30 days after ischemia [68]. However, in the medial temporal lobe cortex, autophagy gene overexpression was observed during 30 days of reperfusion after a reversible cerebral ischemic episode [63,64,65]. This overexpression was accompanied by the downregulation of *caspase 3* gene 2 days after brain ischemia [63,64,65]. Next, on days 7 and 30 after ischemia, the above gene was impressively upregulated [63,64,65]. Thus, the presented alterations indicated that the dysregulation of the expression of autophagy and apoptotic genes may be associated with a different response of neurons in the CA1 and CA3 areas of the hippocampus and in the medial temporal lobe cortex to transient global brain ischemia [63,64,65,66,68].

## 5. Autophagy and Neuronal Death in Ischemic Brain Injury

Although autophagy induction is very important following ischemic brain injury, it is still not the only one mechanism associated with neuronal death. Necrosis and apoptosis are two other forms of neuronal death with big differences in the mechanism and morphology involved. Necrosis, recently called necroapoptosis or necroptosis, is characterized by the swelling and interruption of the cytoplasm membranes [90]. Neurons undergo necrosis in a programmed fashion as ordered cellular explosions [90]. Apoptosis is a programmed neuronal cell death characterized by membrane blabbing, an apoptotic body formation, mitochondrial membrane damage, nuclear condensation, cell shrinkage and DNA fragmentation [91]. Autophagy, apoptosis and necroptosis are morphologically and mechanistically different phenomena, but there is important crosstalk that occurs between them in brain ischemia injury [91]. The autophagy system shares mutual molecular mediators with apoptosis and necroptosis, such as Bcl-2, AMPK and p62 [92]. These integrative hubs of cell signaling, membrane trafficking and physiology also regulate both protein complex formation and the metabolic status sensing of cells, as well as membrane trafficking in autophagy, apoptosis and necroptosis phenomena through control signaling transduction [91,92].

## 6. Crosstalk between Autophagy, Necroptosis and Apoptosis after Brain Ischemia

In recent years, the research of understanding neuronal cell death in brain ischemia injury has transformed into necroptosis and autophagy machinery, combined with the understanding of apoptosis phenomena [90,91,92]. The autophagy process controls neuronal death switching betwixt apoptosis and necroptosis [93]. Neural cell death can be affected by many factors, such as age, gender and duration of ischemia. Autophagy is especially severe in adult brains with ischemic injury [94]. It has also been shown that following ischemic brain injury, female brain neuronal cells displayed a stronger activity of caspase 3 compared to the male brain [95]. The rapidity and intensity of autophagy machinery in brain ischemia are not the same in miscellaneous brain areas, e.g., in the CA1 and CA3 areas of the hippocampus and the temporal lobe of the brain [63,64,65,66,68]. Moreover, the autophagy machinery in different brain areas seems to be related to different forms of neural death [67]. In ischemic neurons in the brain cortex, both autophagy and apoptosis phenomena occurred following ischemic brain injury [96,97]. The hippocampus neurons in the CA1 area died in a strong apoptotic way with only a slightly enhanced autophagy, while neurons in the CA3 sector of the hippocampus experienced a more pure autophagic neuronal death [96]. Another study complied with the above observations and demonstrated that autophagy was only induced in the CA3 area of the hippocampus, but not in the CA1 subfield, following ischemia [65,66,72]. In an animal model of global brain ischemia, the overexpression of the autophagy gene in the CA1 region of the hippocampus was not noted, but overexpression was found in the CA3 area and temporal lobe, indicating neuroprotection by autophagy neurons in both structures against ischemic injury [63,64,65,66,68]. On the contrary, necrosis and apoptosis processes, including plasma membrane burst, DNA fragmentation, caspase 3 gene overexpression, protein activation and AIF activity, were dominated in hippocampus CA1 neurons at 24–48 h following brain ischemia [65,66,72]. Thus, the preference for autophagy machinery activation in the CA3 area of the hippocampus and cortical neurons could explain why the CA3 sector of the hippocampus and cortical neurons were more resistant to brain ischemia compared with the pyramidal neurons in the CA1 area of the hippocampus [38,98].

Autophagy is an adaptive mechanism and probably influences the disintegration and clearance of damaged organelles and/or proteins [61,72,87,99,100,101,102,103,104]. The literature on this issue contains contradictory data concerning autophagy (Table 1). Some studies showed the commitment of autophagy in cell death because of the excessive degradation and clearance of cellular organelles and proteins [67,71,77,105,106,107,108]. Other investigations further pointed out that autophagy machinery probably protects neurons from apoptosis [67,77,103]. The underling mechanisms of necroptosis (programmed necrosis) or the necrosis of neurons after brain ischemia are still not fully understood. The above process was especially severe in ischemic neurons in the CA1 region of the hippocampus compared with the CA3 area [109].

In the first study of necroptosis after brain ischemia, the necroptosis inhibitor necrostatin-1 was unable to block autophagy [133]. Considering that necrostatin-1 reduced the volume of brain infarct, it could be concluded that necroptosis contributes to neural damage after ischemia via a machinery different from that of apoptosis, and necroptosis and autophagy are activated independent of each other during neuronal death [133]. In a recent piece of research, it was observed that the receptor-interacting protein 1 kinase regulated necroptosis-activated autophagy–lysosome machinery, thus, leading to ischemic neurons and astrocyte cell death in focal brain ischemia [134].

Parthanatos, a unique condition of neuronal death, was presented in 2008 [135]. Parthanatos occurs during the overactivation of the nuclear enzyme poly (ADP-ribose) polymerase-1. It was recently presented that the use of inhibitors of parthanatos, but not those against autophagy and/or apoptosis, caused a decrease in neuronal death following hemorrhagic brain ischemia [136]. Notwithstanding, the relationship between parthanatos and autophagy during brain ischemia–reperfusion injury remains unresolved even now. The deletion of the autophagy gene Atg5 diminished the SIRT1 activity important for parthanatos, while parthanatos could not be influenced by the autophagy inhibitor BAF [137]. This suggested that parthanatos might have no association with autophagy. However, the number of reports on the crosstalk between autophagy and necroptosis has been relatively small thus far, and further studies are in demand on the exploration of this issue.

## 7. Autophagy versus Postischemic Brain Injury

There is evidence in the literature that the autophagy gene is dysregulated in the postischemic brain [63,64,65,66,68]. Currently, two opposing activities of autophagy have been suggested, i.e., neuroprotection and influence on neuronal cell death (Table 1). Some in vivo studies have demonstrated the protective functions of autophagy in brain ischemia (Table 1) [102,103], while other studies have presented the detrimental effects of autophagy in ischemia–reperfusion brain injury (Table 1) [108,112,118,134].

### 7.1. The Protective Role of Autophagy in Postischemic Brain

Some studies have presented that autophagy may have a neuroprotective influence in postischemic brain neurodegeneration. In global brain ischemia in rats, the administration of the autophagy activator rapamycin decreased neuronal death in the hippocampal CA1 area and coincided with an increase in markers of autophagy, such as beclin 1 [110]. A single dose of metformin initiated autophagy by activating AMPK in permanent focal brain ischemia in rats, thus, exerting a protective role [87]. Metformin reduced infarct size, neurological deficits and neuronal apoptosis in postischemic brain injury [87]. The neuroprotective action of metformin was fully abolished by compound C and partially by 3-methyladenine [87]. After transient brain ischemia in rats, the administration of isoflurane significantly improved the cognitive and memory functions and coincided with an increase in markers of autophagy, such as beclin 1, and further inhibited the release of inflammatory factors [116]. In immune-related GTPase M1 knockout mice, severe brain damage was reported in a postischemic episode and decrease in the activity of autophagy than in wild-type mice [115]. In addition, the infarct volume in these mice after permanent local brain ischemia significantly increased compared to that in wild-type mice [115]. Neuropathological studies suggested that, during 24 h of recirculation, the immune-related GTPase M1-dependent autophagic response is connected through protective action on neurons from necrosis in the core of infarct and support of neuronal apoptosis in the penumbra [115]. Other genetic models of mice, such as GPR30 [113] and ARRB1 knockout mice [77] and ATF6 knockin mice [114], presented that autophagy had a protective role in postischemic brain neurodegeneration. After global ischemia, hamartin conferred neuroprotection against ischemia by inducing autophagy and increasing locomotor activity [72]. It was presented that ischemic preconditioning through the activation of AMPK induced autophagy activity in postischemic brain, which occurred with a reduction in infract size, neurological deficits and neuronal apoptosis after permanent focal brain ischemia [112]. Ischemic preconditioning-induced autophagy and autophagy could be abolished by compound C or 3-methyladenine [112]. Next, pretreatment with eugenol attenuated ischemia–reperfusion brain injury in rats by inducing autophagy via the AMPK/mTOR/P70S6K signaling pathway [111]. Another study indicated that schaftoside blocked apoptosis and neuroinflammation, reducing brain edema and improving neurologic deficits by enhancing autophagy after transient focal brain ischemia in rats [117].

Many studies did not take sex as a variable when analyzing the experimental outcomes after brain ischemia. However, accumulating data show that sex differences play an important role in the regulation of the autophagy pathway, and may lead to different outcomes between males and females [138,139,140]. Genes located on the X chromosome are thought to possibly explain the sex differences between menopausal men and women in morbidity, mortality and disability rates after an ischemic cerebral event. Although the exact role of autophagy in the pathogenesis of postischemic brain is ambiguous and inconsistent, it is unanimously believed that a moderate activation of autophagy is neuroprotective, while excessive autophagy activation is harmful [141]. The inconsistent results of these studies may be due to several reasons. First, the time of the activation or inhibition of autophagy was different among these studies. As a self-protective mechanism, the activation of autophagy at the early stage of brain ischemia is neuroprotective via degrading misfolded proteins and damaged organelles to maintain the intracellular environment [142]. Interestingly, several studies suggested that autophagy participates in ischemic preconditioning-inducing neuroprotection [74,112,143,144]. Therefore, the activation of autophagy before or during ischemia is neuroprotective. However, it is known that ischemic stress also continues during recirculation, therefore, autophagy is constantly activated, causing further damage. Second, the extent of autophagy activation varied among these studies due to the different models and modulators. As mentioned before, moderate autophagy activation is protective, while excessive autophagy activation is deleterious in the ischemic brain. Last, but not least, the molecules used in some studies were not specific to autophagy. For example, 3-methyladenine is widely used as an inhibitor for autophagy in different studies. However, 3-methyladenine is not specific for targeting autophagy signaling. 3-methyladenine acts as an autophagy inhibitor [145] by inhibiting the PI3K pathway, which participates in other biological processes, including necrosis and apoptosis [121,146], besides autophagy activation [147,148]. Rapamycin, an autophagy activator, initiates autophagy via inhibiting mTOR signaling. However, the mTOR pathway is also suggested to have immunosuppressive and antiproliferative impacts [149,150]. Therefore, it is hard to tell which pathway caused by these molecules actually influences the outcome of postischemic brain. To solve these problems, standardized protocols of brain ischemia models and more specific molecules for autophagy need future studies.

### 7.2. The Deleterious Role of Autophagy in Postischemic Brain

Studies in experimental brain ischemia suggest that autophagy may exhibit deleterious effects on postischemic outcome (Table 1). Autophagy inhibitors, wortmannin and 3-methyladenine, decreased infarction size in rats after permanent local brain ischemia [108]. Presenting data suggest that the inhibition of autophagy blocks the cathepsin–tBid–mitochondrial apoptotic pathway through the stabilization of lysosomal membranes, possibly due to the upregulation of the lysosomal Hsp70.1B in postischemic cells [108]. Autophagy, which was activated in postischemic astrocytes, mildly decreased cell survival after focal permanent brain ischemia [79]. The Beclin 1 knockdown gene of autophagy could prevent secondary thalamic injury after middle cerebral artery occlusion in rats [122]. Another study presented that neuronal injury in permanent local brain ischemia was associated with autophagy and lysosomal signaling [123]. Blocking autophagy ameliorated neurological deficits after focal brain ischemia in estradiol-deficient mice [151]. A recent study suggested that TIGAR protected against neuronal injury partly by inhibiting autophagy [132]. Interleukin-21 was shown to promote neuronal injury and autophagy activity in neurons postischemia, suggesting that autophagy may exert a harmful impact on focal brain ischemia [129]. GPR37 knockout mice showed increased infarct volume and autophagic neuronal death compared with wild-type mice postischemia, which suggested that autophagy participated in neurodegeneration after ischemia and its consequence was damaging [131]. The administration of melatonin before ischemia was shown to protect acute neuronal cell damage postischemia by inhibiting endoplasmic reticulum stress-dependent autophagy via PERK and IRE1 signaling [118]. Other data presented the neuroprotective effect of carnosine, which partially mediated a positive effect through the mitochondria protection and attenuation of negative autophagy processes [119]. The bafilomycin protection of neuronal damage in rat permanent local brain ischemia has been associated with the inhibition of autophagy and lysosomal pathways [123]. Next, a study found that homocysteine may trigger excessive autophagy, thereby facilitating the toxicity of homocysteine on neural stem cells in postischemic brain injury [120]. Another substance, silibinin, protected neuronal cells by inhibiting both the mitochondrial and autophagic cell death pathways after focal brain ischemia in mice [125]. N-acetyl-serotonin offered neuroprotection by inhibiting mitochondrial death signaling and autophagy activation in focal brain ischemia in mice [105]. Other interesting substances such as schizandrin inhibit autophagy through the regulation of AMPK–mTOR signaling, and may have neuroprotective value for ischemic neurons [126]. After the administration of lithium in an experimental model of hypoxia–ischemia, pathology in the hippocampus, cortex, striatum and thalamus was reduced. Lithium reduced the dephosphorylation of glycogen synthase kinase-3β and extracellular signal-regulated kinase, the activation of caspase 3 and the apoptosis-inducing factor, as well as autophagy [130].

## 8. Potential Therapeutic Strategies for Autophagy Modulation Postischemia

According to the aforementioned studies, therapeutic strategies targeting autophagy modulation may be a possible approach in the management of the postischemic neurodegeneration of Alzheimer’s disease proteinopathy. To date, multiple potential therapeutic molecules have been explored [152,153,154,155,156]. These molecules influence different processes of autophagy, including inducing adaptive autophagy and inhibiting excessive autophagy after brain ischemia (Table 2). A variety of compounds have been shown to induce adaptive autophagy. Among these molecules, rapamycin has been widely examined in the management of ischemic brain injury via the inhibition of mTOR. Recent studies have indicated that the administration of rapamycin in rodents undergoing focal brain ischemia could diminish infarct volume, reduce neuronal injury and improve neurological recovery [157,158,159]. Rapamycin has also been reported to reduce endothelial cell death and protect blood–brain barrier permeability in local brain ischemia [160]. A recent review including 17 publications demonstrated that rapamycin significantly decreased infarct size by 22% and improved neuroscores by 31% [159]. Interestingly, lower doses of rapamycin showed greater efficacy at reducing infarct size than higher doses, which was potentially due to an optimal level of autophagy activation being reached with a low dose of rapamycin [159,161,162]. Resveratrol, a common dietary polyphenol, has been shown to extend the clinical therapeutic window of r-tPA for stroke patients [163]. He et al. revealed that resveratrol alleviated ischemia–reperfusion brain injury and reduced infarct volume [164], which was consistent with another study [165]. For the inhibition of excessive autophagy, several studies demonstrated that dexmedetomidine was capable of rendering neuroprotection in focal brain ischemia via the inhibition of excessive autophagy. In the transient middle cerebral artery occlusion, dexmedetomidine protected a mouse brain from ischemia–reperfusion injury via the inhibition of neuronal autophagy through the upregulation of HIF-1*α* [87]. Moreover, dexmedetomidine has been reported to reduce the autophagy effect and improve learning and memory in a rodent model of local brain ischemia [166]. A recent study showed that the regulation of miR-199 was a potential mechanism by which dexmedetomidine inhibited autophagy and promoted neurological outcome postischemia [167]. Propofol administration decreased the infarct volume and improved the neurological outcome after acute focal brain ischemia [168,169]. Recent studies presented that propofol protected against ischemia–reperfusion brain neurodegeneration through the inhibition of excessive autophagy through the regulation of mTOR/S6K1 or long noncoding RNA SNHG14 [170,171]. Melatonin was found to significantly alleviate the consequences of brain insult, such as neuronal apoptosis and brain edema postischemia, through the inhibition of endoplasmic reticulum stress-induced excessive autophagy [118]. Recently, Gao et al. showed that icariside II protected neurons in a rodent model of focal brain ischemia by inhibiting excessive autophagy through interfering with PKG/GSK-3*β* signaling [172]. Although accumulating evidence has shown that molecules targeting autophagy signaling have neuroprotective potential for the consequences of cerebral ischemia, there are several limitations in need of consideration. Firstly, the possible side effects of the agents should be considered. Besides enhancing autophagy, mTORC1 inhibition blocks nucleotide and protein synthesis, inhibiting metabolism and cell proliferation [173]. Long-term rapamycin administration may cause immunosuppression, and glucose intolerance due to mTORC1 is acutely sensitive to rapamycin, whereas mTORC2 is chronically sensitive to rapamycin in vivo [174]. In addition, current research mainly focuses on the effects of autophagy regulation on neuronal cell damage, and less on cell growth after ischemia. Finally, the end result of autophagy-targeted treatments is related not only to the severity of autophagy, but also to the time of the administration of autophagy regulators, the dose of the drug and the route of administration.

## 9. Conclusions

The data indicate that autophagy plays an important role in the control of homeostasis in brain tissue by removing and recycling redundant cell elements and misfolded proteins after cerebral ischemia, thereby regulating the survival and death of neuronal cells (Table 2). Some studies have shown the protective properties of autophagy in cerebral ischemia, while other studies have shown completely opposite properties (Table 1). Thus, autophagy is now presented as a double-edged sword with possible therapeutic potential in cerebral ischemia (Table 1). As such, the scientific community must reach a consensus in the near future on the exact role of autophagy in ischemic brain injury. This review presented the latest data linking postischemic autophagy to Alzheimer’s disease and the role of the ischemic regulation of autophagy in the development of full-blown Alzheimer’s disease. Experimental and clinical evidence shows ischemic brain injury and Alzheimer’s disease are not just traveling companions, but partners in crime. Regardless of the complexity and variety of experimental models, such as cell and animal models, ischemia and reperfusion duration and differences in species, sex and age of animals can cause the observed discrepancies in studies. The data indicate that the physiologically regulated process of autophagy can increase neuronal viability in an emergency, while exaggerated or prolonged ischemia-induced autophagy can be lethal. Some studies indicate that the neuroprotective effect of autophagy is related to the duration of cerebral ischemia, with the prolongation of ischemic time resulting in neurotoxic effects (Table 1). Another explanation for the bidirectional action of autophagy can be related separately to ischemia and the subsequent neuropathology during reperfusion. Thus, autophagy may be neuroprotective during cerebral ischemia but detrimental during recirculation. Conclusions based on current research are preliminary and do not conclusively clarify whether autophagy is a friend or foe in cerebral ischemia. It should also be taken into account that the use of genetic methods also has limitations due to the nonautophagic roles of autophagy-related proteins and the behavior of the *BNCE 1* gene, which had different expressions in different brain structures after ischemia [63,64,65,66,68]. Regarding the above observation, further research is needed to understand what causes the differences in neuronal autophagy in different areas of the brain. In summary, the processes underlying the elimination/recycling of proteins associated with neurodegeneration are currently completely unclear.

Autophagy is widely believed to be a self-protecting cellular catabolic pathway through which some long-lived or misfolded proteins and damaged organelles are degraded and circulated to maintain cellular homeostasis [179]. Many studies have demonstrated that autophagy protects cells from death by inhibiting apoptosis. In addition to preventing apoptosis to inhibit cell death, autophagy also causes excessive cell death. Therefore, autophagy is called a type II programmed cell death to distinguish it from type I programmed cell death apoptosis. Whether autophagy is beneficial or deleterious depends on the rate of autophagy induction and the duration of autophagy activation. Recent evidence has shown that autophagy is activated in various cell types in the brain, such as neuronal, glial and brain microvascular cells, upon ischemic brain injury. Under these conditions, autophagy participates in the regulation of neuroinflammation and damage to neurons and cells of cerebral vessels and surrounding tissues, and this regulation may be positive after ischemia. It can also have a negative effect, and this regulation may be related to the speed, range and damage of the cell types of the cerebrovascular network. However, the specific role of autophagy in the development and progression of cerebrovascular diseases, as well as when and how autophagy itself can regulate autophagy’s own regulators to play a protective role in the process of cerebrovascular diseases, is still unclear.

## 10. Good or Bad Autophagy: A Matter of Balance?

The controversy surrounding autophagy after cerebral ischemia raises the question of whether autophagy is good or bad (Table 1). It is probably more appropriate to view autophagy as a balancing act (Figure 1). Taking into account both experimental and clinical observations, autophagy is essential during reperfusion to remove dysfunctional cytoplasmic organelles or convert pathologically altered proteins, e.g., into energy. On the other hand, excessive autophagy may not meet the cell’s requirements and may lead to cell death (Figure 1). In the second extreme case, with minimal autophagy activity during cell starvation, it is not possible to maintain cellular homeostasis. Therefore, the problem is still debatable because the golden mean has not been found.

As mentioned above, accumulating evidence indicates that autophagy plays a key role in the process of brain neurodegeneration after ischemia, indicating a potential therapeutic target in cerebral ischemia. Currently, however, there are many unanswered questions that need to be thoroughly and critically addressed in future research to translate autophagy-based cerebral ischemia therapies into clinical practice. First, are there noncanonical pathways that initiate nonadaptive autophagy that is detrimental to neuronal cell survival? Postischemic autophagy involves a variety of pathways, but which mechanisms regulate the degree of autophagy is not clear. Excessive autophagy is characterized by the accumulation of autophagosomes, but the mechanisms underlying this phenomenon are still unclear. How can cell-specific autophagy be selectively modulated without activating undesirable signaling pathways leading to cell death? Given that cerebral ischemia-induced autophagy has both beneficial and detrimental effects, consideration should be given to investigating the most likely influencing factors, such as finding the optimal time point for autophagy manipulation. Finally, the last and most important point is that the transfer of this therapeutic strategy from the laboratory to the clinic should be accompanied by robust preclinical studies in appropriate cell cultures and animal models.

## 11. Challenges

The quality control of the cellular cytoplasm of neuronal and glial cells is important for homeostasis, and an uninterrupted and constant balance between various cellular organelles and proteins is required. It has been suggested that changes in the quality of the cytoplasm are involved in the pathogenesis of cerebral ischemia, leading to the death of neuronal and other brain cells. The modulation of molecular processes related to autophagy after ischemia, which was shown in the review, creates opportunities for designing new therapies. However, to take advantage of this potential opportunity, a better understanding of the mechanism of autophagy is required with the development of new tools, such as transgenic animal models, to study this complex phenomenon.

## Figures and Tables

**Figure 1 ijms-24-13793-f001:**
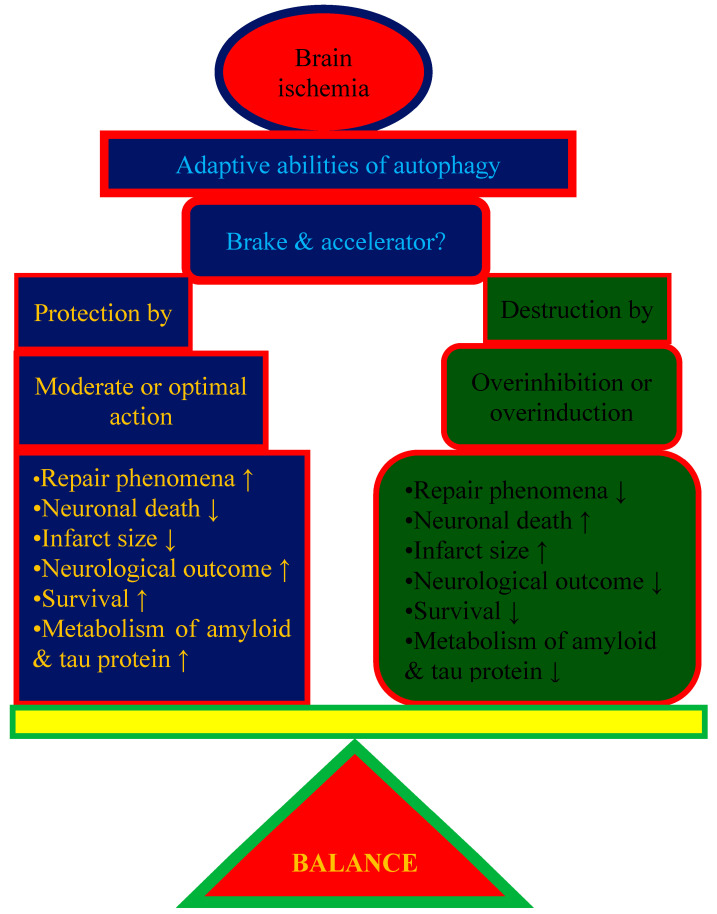
There is a delicate balance between the beneficial and harmful effects of autophagy in postischemic brain neurodegeneration. ↑—increase; ↓—decrease.

**Table 1 ijms-24-13793-t001:** The role of autophagy in postischemic brain neurodegeneration.

Reference	Model	Animal	Autophagy Induction by	Autophagy Inhibition by	Role
[72]	4VO	Rats	Tsc1 induction		Protective
[110]	4VO	Rats	Rapamycin induction		Protective
[111]	tMCAO	Rats	Eugenol induction		Protective
[87]	pMCAO	Rats	Metformin induction		Protective
[112]	pMCAO	Rats		Compound C inhibition	Protective
[113]	tMCAO	Mice		GPR30 knockout inhibition	Protective
[77]	tMCAO	Mice		ARRB1 knockout inhibition	Protective
[114]	tMCAO	Mice	ATF6 knockin induction		Protective
[115]	pMCAO	Mice		Immune-related GTPase M1 knockout inhibition	Protective
[116]	tMCAO	Rats	Isoflurane induction		Protective
[117]	tMCAO	Rats	Schaftoside induction		Protective
[118]	tMCAO	Rats		Melatonin inhibition	Harmful
[108]	pMCAO	Rats		3-methyladenine or wortmannin inhibition	Harmful
[119]	tMCAO or pMCAO	Rats		Carnosine inhibition	Harmful
[120]	tMCAO	Rats	Homocysteine induction		Harmful
[121]	pMCAO	Rats		3-methyladenine inhibition	Harmful
[122]	tMCAO	Rats		Becn1-shRNA or 3-methyladenine inhibition	Harmful
[123]	pMCAO	Rats		Bafliomycin A1 or 3-methyladenine inhibition	Harmful
[124]	tMCAO	Mice	IL-17A induction		Harmful
[105]	pMCAO	Mice		N-acetyl-serotonin inhibition	Harmful
[125]	pMCAO	Mice		Silibinin inhibition	Harmful
[126]	tMCAO	Mice		Schizandrin inhibition	Harmful
[127]	tMCAO	Mice		3-methyladenine inhibition	Harmful
[128]	tMCAO	Mice		CircHECTD1 knockdown inhibition	Harmful
[129]	tMCAO	Mice		IL-21 knockout inhibition	Harmful
[130]	Hypoxic-ischemic	Neonatal rats		Lithium inhibition	Harmful
[131]	tMCAO	Mice	GPR37 knockout induction		Harmful
[132]	tMCAO	Mice		TIGAR knockout induction and TIGARtransgene inhibition	Harmful

4VO—four-vessel occlusion; tMCAO—transient middle cerebral artery occlusion; pMCAO—permanent middle cerebral artery occlusion.

**Table 2 ijms-24-13793-t002:** Compounds modulating autophagy after brain ischemia.

References	Molecules	Effect on Autophagy	Action by	End Result on Ischemia
[157,158,159,160,161,162,175]	Rapamycin	Induction	mTOR-dependent activity	↓ Infarct volume, neuronal injury, neurological deficits, endothelial cell death, BBB injury
[163,164,165]	Resveratrol	Induction	Sirt1-dependent autophagy activity	↓ Infarct volume, inflammation, brain edema, apoptosis, neurological deficits. ↑ Therapeutic window
[166,167,176]	Dexmedetomidine	Inhibition-excessive autophagy	Upregulation of HIF-1*α*	↓ Neuronal injury, neurological deficits, beclin 1, caspase 3. ↑ Learning, memory, neurological outcome
[168,169,170,171]	Propofol	Inhibition-excessive autophagy	Regulation of mTOR/S6K1	↓ Infarct volume. ↑ Neurological outcome
[118]	Melatonin	Inhibition-excessive autophagy	Inhibiting ER stress-dependent autophagy	↓ Apoptosis, brain edema, neurological deficits
[172]	Icariside II	Inhibiting-excessive autophagy	PKG/GSK-3*β* signaling	↓ Autophagic neuronal death
[72]	Hamartin	Initiation	Inhibition of mTORC1	↑ Neuroprotective effect
[177]	Bexarotene	Initiation	Inhibition of autophagosome degradation	↓ Infarct size, behavioral deficits
[178]	Melatonin	Inhibition	Activating the PI3K/Akt pro-survival pathway and inhibiting expression of beclin-1	↓ Infarct size, neurological deficits
[73]	Minocycline	Inhibition	Suppressing beclin-1	↓ Cell damage, caspase 3 and 8
[127]	Fingolimod	Suppression	Activation of mTOR/p70S6K pathway, decrease in autophagosome and beclin 1	↓ Infarct volume, neuronal apoptosis, functional deficits
[105]	N-acetyl-serotonin	Suppression	Reducing the activation of autophagy	↓ Mitochondrial cell death pathway, autophagic cell death
[123]	3-methyladenine	Increase in autophagos-omes	Inhibition on ischemia-induced upregulation of LC3-II	↓ Infarct volume, brain edema, motor deficits
[119]	Carnosine	Attenuation	Attenuation of deleterious autophagic process	↑ Improvement in brain mitochondrial function, mitophagy signaling
[87]	Metformin	Initiation	Activation of brain AMPK,	↓ Infarct volume, neurological deficits, cell apoptosis
[175]	Lithium carbonate	Induction	mTOR-independent	↓ Evans blue extravasation, brain water

↑—increase; ↓—decrease.

## Data Availability

Not applicable.

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
