# Peer review of "The Dual Role of Autophagy in Postischemic Brain Neurodegeneration of Alzheimer’s Disease Proteinopathy"

_ijms, 2023, doi:10.3390/ijms241813793_

Round 1

Reviewer 1 Report

In this manuscript. Ryszard Pluta gives an extensive review of the role of autophagy in post-ischemic brain neurodegeneration of Alzheimer's disease proteinopathy. The author explores in an exhaustive manner the ischemic brain injury in humans and the relationship with autophagy. In addition, the author provides a section to illustrate the potential therapeutic strategies for autophagy modulation post-ischemia.

Follow, I will give some suggestions that can help to improve the manuscript.

- Firstly, I suggest author to include a section where describe a general overview of autophagy. This can be useful for a reader that is starting to approach to this molecular mechanism.

- I encourage authors to also well describe mitophagy, since growing evidence demonstrate the involvement of this pathway in Alzheimer (PMID: 30538104). A brief paragraph should also include the dementia status provoked by Alzheimer ( PMID: 36908880, PMID: 31775806)

- I also suggest author to include a brief section to describe the recent discoveries regarding autophagy and mitophagy, in most common neurodegenerative diseases such as multiple sclerosis (PMID: 34099564, PMID: 28866627), Parkinson’s (PMID: 34340748), ALS (PMID: 23023293, PMID: 30126943). This will permit to highlight the importance of autophagy in neurodegeneration.

- Finally, I encourage author to include some figures to summarize the key points of the manuscript.

Dear Authors, I am not a native English speaker.

Therefore I am not totally qualified to judge your English form.

In my opinion, you should only correct some minor errors related to the English style and grammar.

Author Response

Review 1.

All changes are in red in MS.

In this manuscript. Ryszard Pluta gives an extensive review of the role of autophagy in post-ischemic brain neurodegeneration of Alzheimer's disease proteinopathy. The author explores in an exhaustive manner the ischemic brain injury in humans and the relationship with autophagy. In addition, the author provides a section to illustrate the potential therapeutic strategies for autophagy modulation post-ischemia.

Follow, I will give some suggestions that can help to improve the manuscript.

Thanks.

- Firstly, I suggest author to include a section where describe a general overview of autophagy. This can be useful for a reader that is starting to approach to this molecular mechanism.

Done.

- I encourage authors to also well describe mitophagy, since growing evidence demonstrate the involvement of this pathway in Alzheimer (PMID: 30538104). A brief paragraph should also include the dementia status provoked by Alzheimer ( PMID: 36908880, PMID: 31775806)

I'm very sorry. I know and I have the data, but these issues are not the focus of this review.

- I also suggest author to include a brief section to describe the recent discoveries regarding autophagy and mitophagy, in most common neurodegenerative diseases such as multiple sclerosis (PMID: 34099564, PMID: 28866627), Parkinson’s (PMID: 34340748), ALS (PMID: 23023293, PMID: 30126943). This will permit to highlight the importance of autophagy in neurodegeneration.

I'm very sorry. These diseases are not the focus of this review.

- Finally, I encourage author to include some figures to summarize the key points of the manuscript.

Done.

Reviewer 2 Report

In this review, the authors provided an overview of the dual role of autophagy in post-ischemic brain neurodegeneration of AD proteinopathy. The review highlighted the complex relationship between autophagy and neurodegenerative diseases, such as cerebral ischemia and AD. Although autophagy can be neuroprotective by removing toxic protein aggregates and damaged organelles, it can also contribute to neurodegeneration by promoting the accumulation of misfolded proteins and disrupting cellular homeostasis. Overall, this review provided a comprehensive overview of the role of autophagy in post-ischemic brain neurodegeneration of AD. However, I think the authors' case for potential therapeutic applications of autophagy needs to be strengthened, especially since its complex relationship with neurodegenerative diseases must be clearly unraveled. There are also some suggestions provided as consideration for further modifications.

1. The content of the Introduction is very lengthy. I think it is necessary to shorten it a little and focus only on the essential parts. In particular, most of the content mentions ischemia stroke, but the causal relationship between the former and AD does not seem to be very clear. Since the title is such a description, I think it is necessary to further strengthen this.

2. Tab 1: Putting 'Autophagy induction or inhibition' in the same column appears likely to cause confusion. This makes it unclear whether activating (or inhibiting) autophagy is a benefit or a harm. Therefore, I suggest that the authors make appropriate modifications to this manuscript.

3. The authors describe the relationship between autophagy and neurons in great detail. However, we all know that most cell types in the brain are glial cells, and I think it also plays a role in AD. I don’t know what the role of autophagy is in glial cells (especially immune-related activities). The authors may consider adding some content to discuss it.

4. Most of the evidence described in the manuscript is based on the results of animal experiments. Are there any experimental human data to support this?

5. It would be better if the authors could add a scheme figure to illustrate the benefits (or disadvantages) of regulating various autophagy-related factors on neurons under disease conditions.

Only a very small number of English rhetorical or spelling needs to be corrected.

Author Response

Review 2

All changes are in red in MS.

In this review, the authors provided an overview of the dual role of autophagy in post-ischemic brain neurodegeneration of AD proteinopathy. The review highlighted the complex relationship between autophagy and neurodegenerative diseases, such as cerebral ischemia and AD. Although autophagy can be neuroprotective by removing toxic protein aggregates and damaged organelles, it can also contribute to neurodegeneration by promoting the accumulation of misfolded proteins and disrupting cellular homeostasis. Overall, this review provided a comprehensive overview of the role of autophagy in post-ischemic brain neurodegeneration of AD. However, I think the authors' case for potential therapeutic applications of autophagy needs to be strengthened, especially since its complex relationship with neurodegenerative diseases must be clearly unraveled. There are also some suggestions provided as consideration for further modifications.

Thanks.

  1. The content of the Introduction is very lengthy. I think it is necessary to shorten it a little and focus only on the essential parts. In particular, most of the content mentions ischemia stroke, but the causal relationship between the former and AD does not seem to be very clear. Since the title is such a description, I think it is necessary to further strengthen this.

Introduction shortened. and rearrangement.

  1. Tab 1: Putting 'Autophagy induction or inhibition' in the same column appears likely to cause confusion. This makes it unclear whether activating (or inhibiting) autophagy is a benefit or a harm. Therefore, I suggest that the authors make appropriate modifications to this manuscript.

Done.

  1. The authors describe the relationship between autophagy and neurons in great detail. However, we all know that most cell types in the brain are glial cells, and I think it also plays a role in AD. I don’t know what the role of autophagy is in glial cells (especially immune-related activities). The authors may consider adding some content to discuss it.

Autophagy is also related to glial cells after ischemia what is presented in manuscript. However, the presented topic has limited knowledge.

  1. Most of the evidence described in the manuscript is based on the results of animal experiments. Are there any experimental human data to support this?

None.

  1. It would be better if the authors could add a scheme figure to illustrate the benefits (or disadvantages) of regulating various autophagy-related factors on neurons under disease conditions.

Done.